# Pervasive transcription read-through promotes aberrant expression of oncogenes and RNA chimeras in renal carcinoma

Ana R Grosso*, Ana P Leite, Sílvia Carvalho, Mafalda R Matos, Filipa B Martins, Alexandra C Vítor, Joana MP Desterro, Maria Carmo-Fonseca, Sérgio F de Almeida*

Instituto de Medicina Molecular, Faculdade de Medicina da Universidade de Lisboa, Lisboa, Portugal

**Abstract** Aberrant expression of cancer genes and non-canonical RNA species is a hallmark of cancer. However, the mechanisms driving such atypical gene expression programs are incompletely understood. Here, our transcriptional profiling of a cohort of 50 primary clear cell renal cell carcinoma (ccRCC) samples from The Cancer Genome Atlas (TCGA) reveals that transcription read-through beyond the termination site is a source of transcriptome diversity in cancer cells. Amongst the genes most frequently mutated in ccRCC, we identified *SETD2* inactivation as a potent enhancer of transcription read-through. We further show that invasion of neighbouring genes and generation of RNA chimeras are functional outcomes of transcription read-through. We identified the *BCL2* oncogene as one of such invaded genes and detected a novel chimera, the *CTSC-RAB38*, in 20% of ccRCC samples. Collectively, our data highlight a novel link between transcription read-through and aberrant expression of oncogenes and chimeric transcripts that is prevalent in cancer.

*For correspondence: agrosso@medicina.ulisboa.pt (ARG); sergioalmeida@fm.ul.pt (SFdA)

Competing interests: The authors declare that no competing interests exist.

## Introduction

Clear cell renal cell carcinoma (ccRCC) is the most common histological subtype of renal carcinoma. The genetics of ccRCC is dominated by either somatic or germline inactivating mutations in the *VHL* gene. Regarding the full spectrum of genomic alterations, ccRCC ranks amongst solid tumors with the lowest average number of point mutations, small indels (*Kandoth et al., 2013*) and somatic copy number alterations (*Zack et al., 2013*). These findings suggest that epigenetic events make a significant contribution for the deregulation of the oncogenic and tumor suppressor gene expression programs that drive ccRCC development and progression. In fact, mutations in ccRCC are frequently observed in epigenetic factors such as the chromatin-remodeler *PBRM1* and the histone modifying enzymes *BAP1* and *SETD2*, highlighting the central role of epigenetic regulation in this particular cancer (*Duns et al., 2010*; *Varela et al., 2011*; *Dalgliesh et al., 2010*; *Creighton et al., 2013*). Such mutations in genes that control the epigenome can strongly modulate the landscape of the tumor transcriptome via aberrant expression of global sets of genes. For instance, defects in transcription termination lead to read-through beyond the annotated 3' gene boundary and have the potential to severely modify the transcriptome and to risk the integrity of vital gene expression programs (*Kuehner et al., 2011*). Paradoxically, the prevalence and functional outcome of transcription read-through has not been thoroughly scrutinized in any cancer before. Here, we report an unprecedented transcriptional profiling of a cohort of 50 pairs of ccRCC tumor and normal matched samples from The Cancer Genome Atlas (TCGA). We show that transcription read-through is prevalent in

**eLife digest** Mutations in genes play important roles in many types of cancer. However, mutations alone cannot explain all the biological changes that occur to cancer cells. For example, very few mutations have been linked with a type of kidney cancer called clear cell renal cell carcinoma (or ccRCC for short). Instead, scientists suspect that this cancer is largely caused by changes in the expression of particular genes so that certain cancer-promoting genes are more highly expressed, while other genes that would prevent tumor growth become less active.

One of the few genes that is often mutated in ccRCC is called *SETD2*. This gene is involved in processes that alter the structure of DNA, but do not alter the genes themselves. These "epigenetic" changes can alter how the instructions in genes are used to make proteins. The first step in making proteins is to use a section of DNA as a template to make molecules of messenger ribonucleic acid (mRNA) in a process called transcription. There are markers within a gene that show where transcription should start and stop to produce the mRNA required to make a particular protein. Epigenetic changes can mask these markers so that the cell produces longer mRNAs that incorporate instructions from neighboring genes.

It was not known how often these stop signs are ignored in ccRCC cells. Here, Grosso et al. compared transcription in normal cells and in ccRCC tumor cells from 50 different patients. The experiments show that more stop signs were ignored in many of the cancer cells, especially in cells with mutations in *SETD2*. This caused all or parts of neighboring genes to be transcribed along with the target gene and led to changes in the expression levels of these genes. For example, a cancer-promoting gene called *BCL2* was more highly expressed in these cells.

Furthermore, some of the mRNA molecules produced in these cancer cells may make "fusion" proteins that combine elements from several proteins. These fusion proteins may work differently to normal cell proteins and therefore might also promote the development of tumors. Grosso et al.'s findings reveal a new link between epigenetic changes and cancer.

ccRCC and found that high levels of transcription read-through correlate with poor survival rates. Amongst the most frequently mutated genes in ccRCC, we identify *SETD2* inactivation as a major driving force of impaired transcription termination and high levels of read-through. Moreover, we show that transcription read-through overruns and interferes with the expression of downstream genes. We identify the anti-apoptotic oncogene *BCL2* as one of such interfered genes, thereby illustrating a new mechanistic basis for the transcriptional deregulation of oncogenes. In addition, our transcriptome analyses revealed recurrent RNA chimeras generated from read-through episodes in ccRCC. RNA chimeras are common features of cancer cells formerly thought to be produced solely by chromosomal translocations. We now know that many chimeric transcripts can originate from DNA-independent events such as *trans*-splicing, RNA recombination or transcription read-through (*Gingeras, 2009*). Our analyses revealed that read-through is a major source of RNA chimeras in ccRCC and identified a novel chimera, the *CTSC-RAB38*, in 20% of ccRCC tumors, but not in any normal matched sample. Altogether, our data disclose the prognostic power of transcription read-through and emphasizes its role as a major source of transcriptome diversity in ccRCC, namely via aberrant expression of cancer genes and RNA chimeras.

## Results

### Transcription read-through is frequent in ccRCC

To investigate the prevalence of transcription read-through in ccRCC, we analysed RNA-seq data from 50 pairs of tumor and normal matched samples from TCGA (*Cancer Genome Atlas Research N, 2013*). Compared to normal tissue, all tumor samples exhibited several genes with transcription termination defects revealed by a high number of reads mapping downstream the transcription termination site (TTS) (*Figure 1A,B* and *Supplementary file 1A*). Such accumulation of reads results from transcription read-through beyond the TTS, a surrogate for deficient transcription termination (*Higgs et al., 1983*). In agreement with a defect in transcription termination, we did not detect

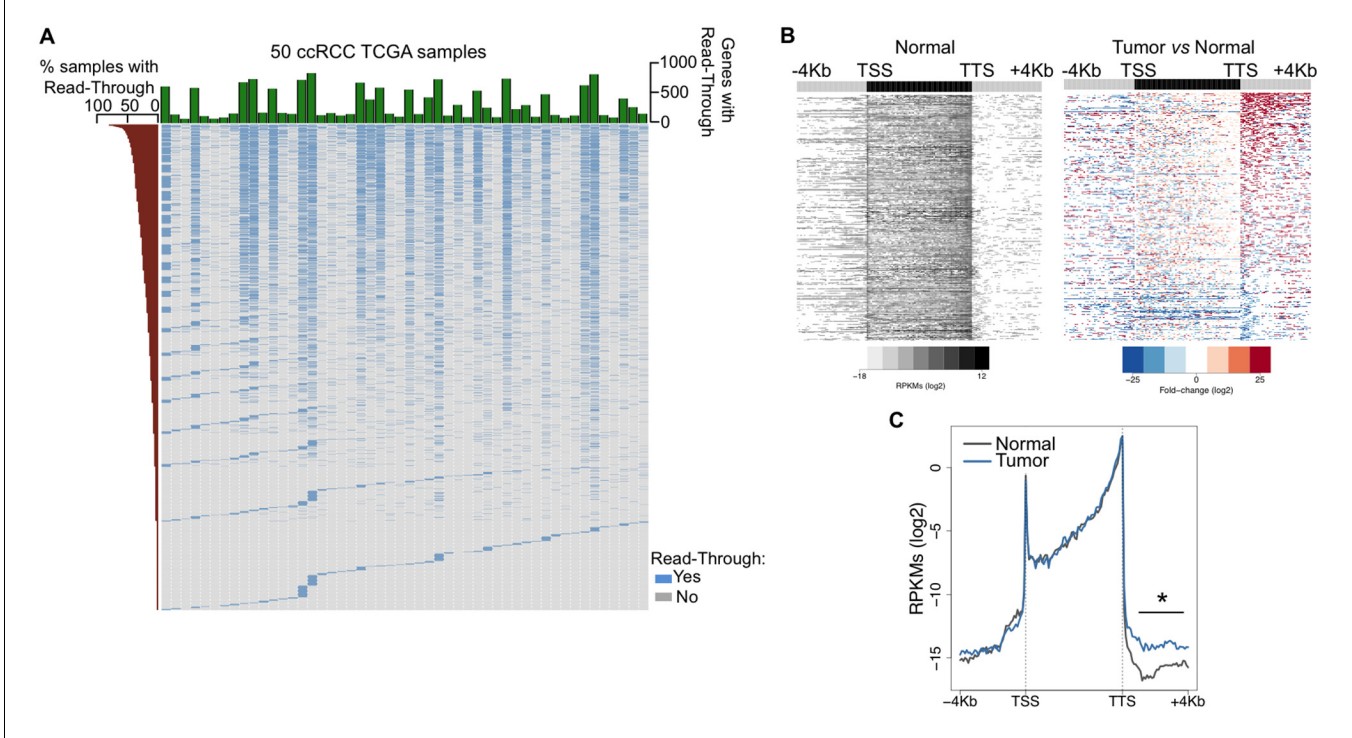

**Figure 1.** Transcription read-through is prevalent in ccRCC. (**A**) Top graph depicts the number of genes with transcription read-through per ccRCC sample. The heatmap illustrates the genes with (blue) and without (grey) transcription read-through. The left graph indicates the percentage of samples on which read-through is observed for each individual gene. (n = 50 tumor/matched normal ccRCC TCGA samples). (**B**) Heatmap representation of the RNA-seq profile distribution and fold change after the TTS region of genes with transcription read-through in one representative TCGA ccRCC sample (patient barcode TCGA-CZ-5465) of a total of 50 tumor and matched pairs analysed. The gene body region was scaled to 60 equally sized bins and ± 4 Kb gene-flanking regions were averaged in 100-bp windows. The left panel shows the read counts (log2 RPKMs) of the matched normal tissue in all genes with read-through and the right panel shows the fold-change (log2) of read counts between the tumor and the matched normal tissue. Genes are ordered according to the read-through length. Scales and colour keys for each panel are depicted in the bottom. (**C**) Metagene analysis of RNA-seq profiles for tumor and matched normal tissue from one ccRCC patient. *p<0.05 by Student's T-test.

differences in read counts on any region upstream the TTS, contrasting with the significant increase in the intergenic region immediately downstream this site (*Figure 1C*).

We then examined whether global deregulation of gene expression at the level of transcription termination affects overall survival rates of ccRCC patients. For that we segregated the TCGA samples into two categories: 'high read-through' samples (those with more than 200 genes with read-through) and 'low read-through' samples (less than 200 genes with read-through) (*Figure 2A*). We found that patients with a 'high read-through' phenotype died significantly earlier than patients with a 'low read-through' phenotype (p = 0.008, log-rank test; *Figure 2B*).

To estimate the contribution of the five most frequently mutated genes in ccRCC (*Cancer Genome Atlas Research N, 2013*) to the observed transcription termination defects, we calculated the percentage of samples carrying any of these mutations that fall within each of the two subsets defined above. Samples with mutations in the histone methyltransferase *SETD2* scored preferentially in the 'high read-through' category (58%) (*Figure 2C*). In contrast, samples carrying any of the four other frequently mutated genes in ccRCC segregated equally between both groups (*BAP1*) or mainly in the 'low read-through' category (*VHL, PBRM1, MTOR. Figure 2C*). Mutations on other genes known to be required for transcription termination or pre-mRNA processing were rare and did not segregate specifically in any category (*Supplementary file 1B*). These data suggest that widespread transcription read-through is a distinctive hallmark of ccRCC and identify *SETD2* mutations as a putative contributing factor for this phenotype.

To further investigate the correlation between *SETD2* mutations and transcription read-through in ccRCC, we performed RNA sequencing of 2 *SETD2* wild type (wt) and 4 *SETD2* mutant ccRCC cell

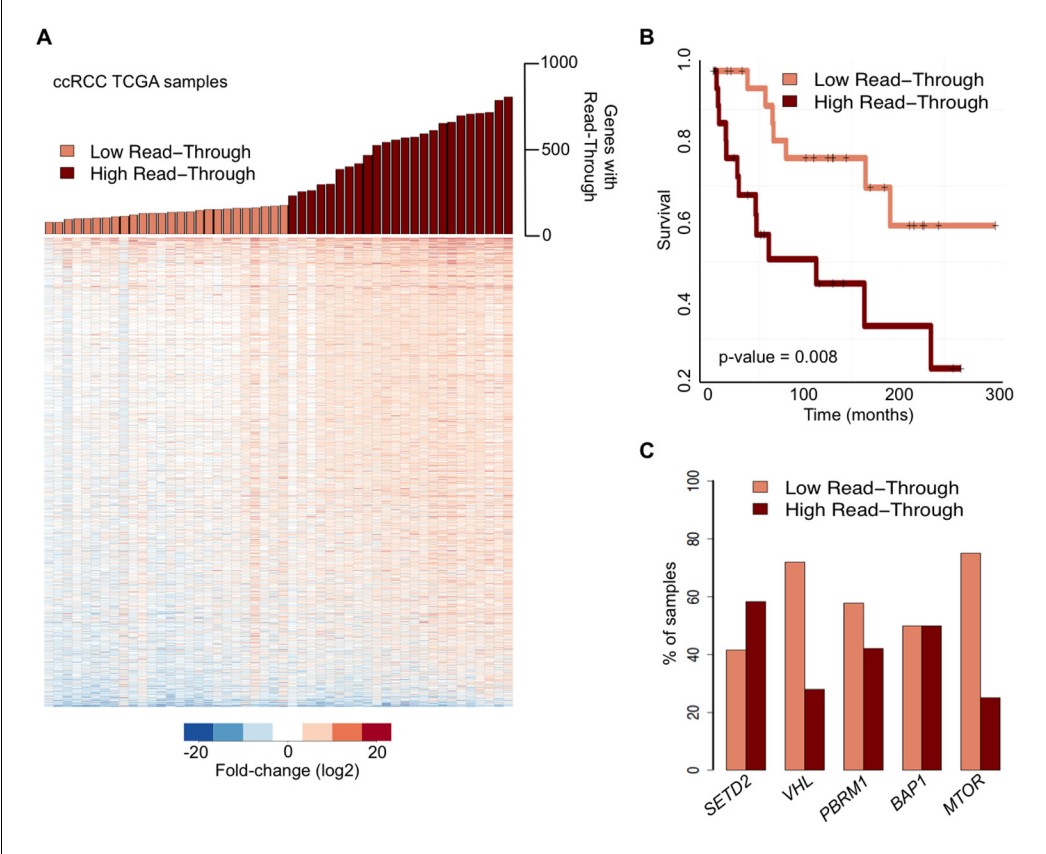

**Figure 2.** Transcription read-through correlates with ccRCC survival rates. (**A**) The top graph indicates the number of genes with transcription read-through on each ccRCC patient sample. Samples were split in two groups according to the number of genes with transcription read-through (low or high), using 200 genes as a cut-off. The heatmap represents the RNA-seq tumor/matched normal fold change 4 Kb after the TTS region of genes with transcription read-through. (**B**) Kaplan-Meier plot comparing the survival of patients separated into 'high read-through' and 'low read-through' subsets as defined in **A**. (**C**) Proportion of ccRCC patient samples with low and high transcription read-through. Results are shown for samples containing any of the most recurrently mutated genes in ccRCC: *SETD2, VHL, PBRM1, BAP1* and *MTOR*. Proportions of high and low read-through were significantly different between samples carrying mutation in *SETD2* and in any of the remaining genes (Fisher's Exact Test p<0.05).

lines previously reported to have a marked reduction of H3K36me3 levels (*Duns et al., 2010*; *Carvalho et al., 2014*). The wt ccRCC cell line Caki2 showed the highest *SETD2* expression levels, comparable with those of a non-cancer kidney epithelial cell line (HEK293) (*Figure 3A*). For this reason, we chose Caki2 as the reference dataset for the pairwise analyses of the remaining wt (Caki1) and mutant (MF, AB, ER, FG2) samples. These analyses revealed transcription read-through in hundreds of genes on all ccRCC cell lines, with a significantly higher incidence in all *SETD2* mutant cells (*Figure 3B* and *Supplementary file 1C*). A metagene analysis of genes with transcription termination defects in *SETD2* mutant cells revealed that expression levels vary significantly downstream the TTS, but not within the gene body or at the promoter region (*Figure 3C*). Moreover, heatmaps of the fold change of read counts between different cell lines depict a strong increase in the number of read-through reads in *SETD2* mutant cells (*Figure 3D* and *Figure 3—figure supplement 1*). To rule out the contribution of different genetic backgrounds in these cell lines, we interrogated RNA-seq data from: *SETD2* knockout (KO) ccRCC cells (*Ho et al., 2015*); *SETD2*-depleted human mesenchymal stem (MS) cells (*Luco et al., 2010*); and embryonic stem (ES) cells from *Setd2* KO mice (*Zhang et al., 2014*). Again, when compared to wt cells, *SETD2*-deficient cells displayed increased read counts specifically in the region immediately downstream the TTS consistent with transcription read-through events (*Figure 3E,F*; *Figure 3—figure supplement 2* and *Supplementary file 1D,E, F*). Similar results were obtained upon RT-qPCR measurements of transcription read-through in three

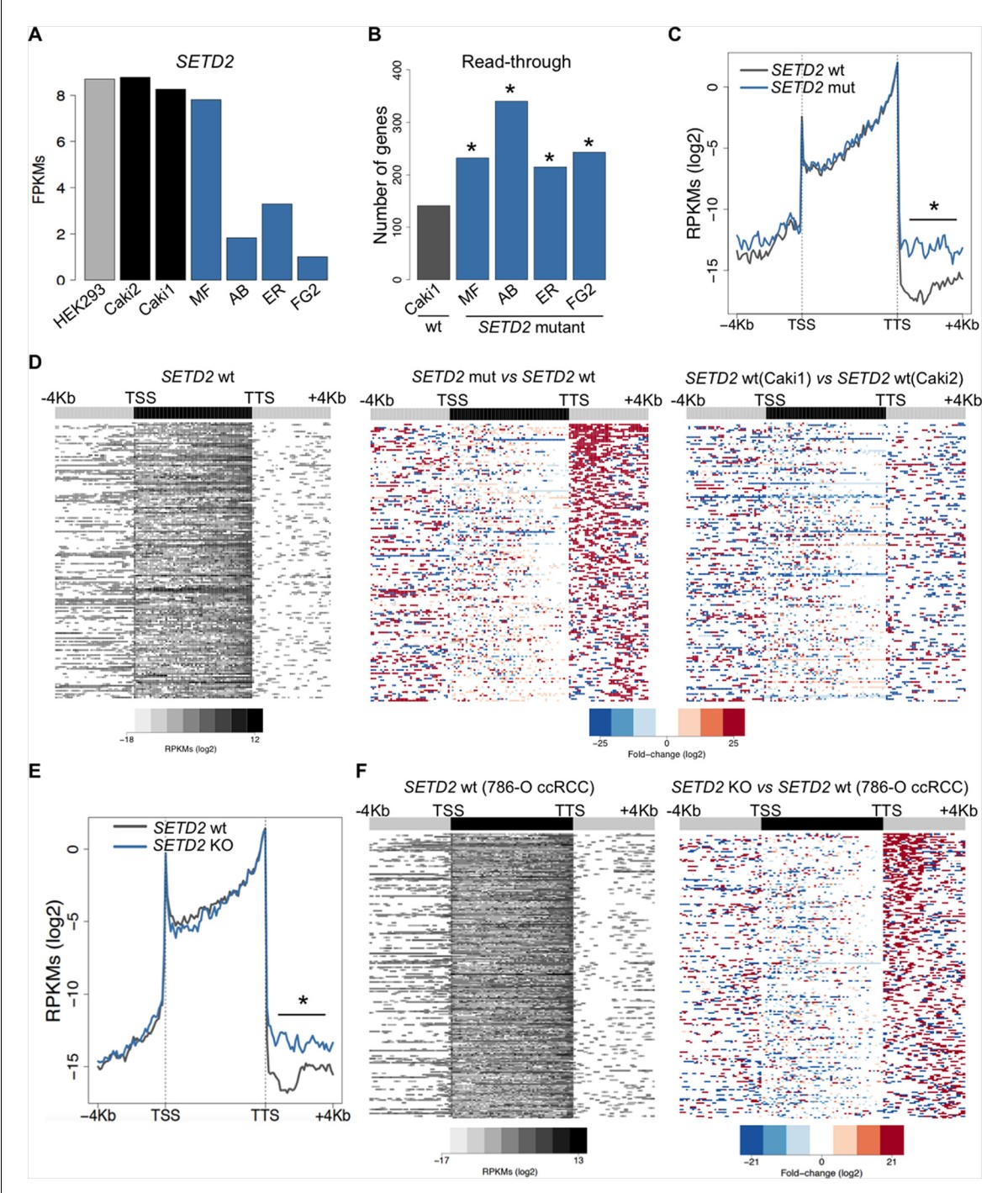

**Figure 3.** SETD2 mutations promote transcription read-through in ccRCC. (A) *SETD2* expression levels (FPKMs) in HEK293 and ccRCC cell lines. (B) Number of genes with transcription read-through up to 4 Kb downstream the TTS. *p<0.05 by Fisher's Exact Test after comparing each *SETD2* mutant cell line with the *SETD2* wt cell line (Caki1); (C) Metagene analysis of genes showing transcription read-through in *SETD2* mutant and wt ccRCC cell lines. The gene body region was scaled to 60 equally sized bins and ± 4 Kb gene-flanking regions were averaged in 100-bp windows. *p<0.05 by Student's T-test; (D) Heatmap representation of RNA-seq profile distribution and fold change after the TTS region of genes showing transcription read-through. Genes were scaled and averaged as in C. The left panel shows the read counts (log2 RPKMs) of the *SETD2* wt cell line (Caki2) in all genes with read-through. The two right panels show the fold-change (log2) between each ccRCC cell line and the reference *SETD2* wt ccRCC cell line (Caki2). Genes are ordered according to the read-through length. Scales and colour keys for each panel are depicted at the bottom of the panel. (E) Metagene analysis (as detailed in C) of genes showing transcription read-through in *SETD2* KO and wt 786-O ccRCC cells. *p<0.05 by Student's T-test. (F) Analysis as described in D of RNA-seq data from *SETD2* KO and control 786-O ccRCC cell lines.

*Figure 3 continued on next page*

*Figure 3 continued*

The following figure supplements are available for figure 3:

**Figure supplement 1.** SETD2 mutations in ccRCC cells promote transcription read-through.

**Figure supplement 2.** SETD2-depletion impairs transcription termination.

**Figure supplement 3.** SETD2-depletion impairs transcription termination in ccRCC cells.

distinct genes 48 hr after depletion of *SETD2* from wt ccRCC cells by RNA interference (***Figure 3— figure supplement 3***)

We then investigated whether wt *SETD2* can rescue the transcription termination defects observed in *SETD2* mutant ccRCC cells. For that, we performed single-molecule RNA FISH after transient expression of GFP-tagged SETD2 in a *SETD2* mutant ccRCC cell line (***Figure 4A***). RNAs produced upon transcription read-through of *MRPL23* and *SEL1L3* were visualized as foci obtained

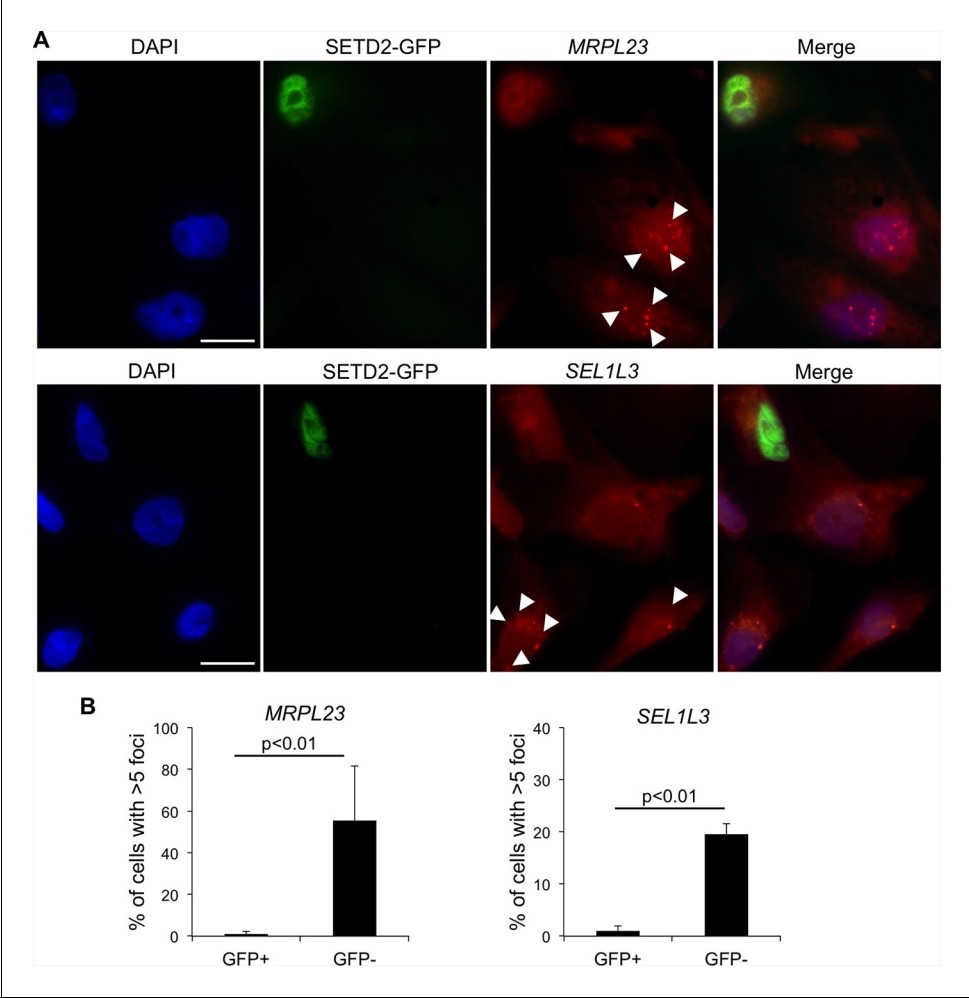

**Figure 4.** SETD2 rescues the transcription termination defects of SETD2 mutant ccRCC cells. (**A**) RNA FISH experiments on a *SETD2*-mutant ccRCC cell line (FG2) transiently expressing wt SETD2-GFP. Quasar570-labeled probes were designed against a region downstream the termination sites of *MRPL23* and *SEL1L3*. The arrows indicate single RNA transcripts generated by a transcription read-through event. (**B**) Quantification of the number of GFP-negative and GFP-positive cells containing more than 5 RNA FISH foci. Means and standard deviations from at least 50 cells from four individual experiments are shown. Scale bars: 10 μm. p<0.01 by Student's T-test.

with RNA FISH probes targeting a region downstream the termination site. These foci were present in significantly higher number in GFP-negative cells than in wt SETD2-GFP expressing cells (*Figure 4B*). This result reveals that expression of wt SETD2 is sufficient to revert the transcription termination defects observed in mutant cells. Altogether, these data support the view that SETD2 is necessary to guide correct transcription termination genome-wide and that mutations in this histone modifier gene cause aberrant transcription patterns in ccRCC.

## Transcription read-through interferes with the expression of neighbouring genes

One possible functional consequence of aberrant transcription read-through is the invasion of adjacent downstream genes altering their expression levels. This *trans*-acting transcriptional interference mechanism may play important roles in cancer development and progression by deregulating the expression of relevant oncogenes and tumor suppressors (*Proudfoot, 1986*). To test whether overrunning of neighbouring genes is a frequent outcome of transcription read-through in ccRCC, we analysed the expression levels of the entire intergenic region and of the gene located immediately downstream. In agreement with our prediction, there was a statistically significant increase in read counts along the intergenic region and within the body of the downstream gene of a tandem pair (*Figure 5A*). This difference was still detected after the TTS of the downstream gene, but not within the upstream gene (*Figure 5B*). Overall, reading-through RNA polymerase II (RNAPII) molecules invaded an average of 20% of genes located downstream (*Supplementary file 1A*).

We then asked if read-through levels correlate with the expression levels of the downstream gene across the TCGA dataset. Notably, expression of 52 out of 903 genes (6%) exhibited a statistically significant correlation with read-through levels of the upstream gene (*Figure 6A*, *Supplementary file 1G*). From these, 51 genes were positively correlated and only one gene showed a negative correlation. Amongst positively correlated genes we found the anti-apoptotic oncogene *BCL2*. In support of its oncogenic role in kidney cancer, depletion of *BCL2* with antisense oligonucleotides inhibits ccRCC tumor growth in vitro and in vivo (*Uchida et al., 2001*). We further observed that expression of *BCL2* is frequently increased in the tumor samples of the TCGA dataset when compared to the matched samples (*Figure 6B*). Importantly, augmented levels of *BCL2* mRNA and protein correlated positively with the levels of transcription read-through of the *KDSR* gene located immediately upstream (*Figure 6B,C*). Altogether, these data suggest that transcription termination defects interfere with the expression of neighbouring genes and illustrate a new paradigm

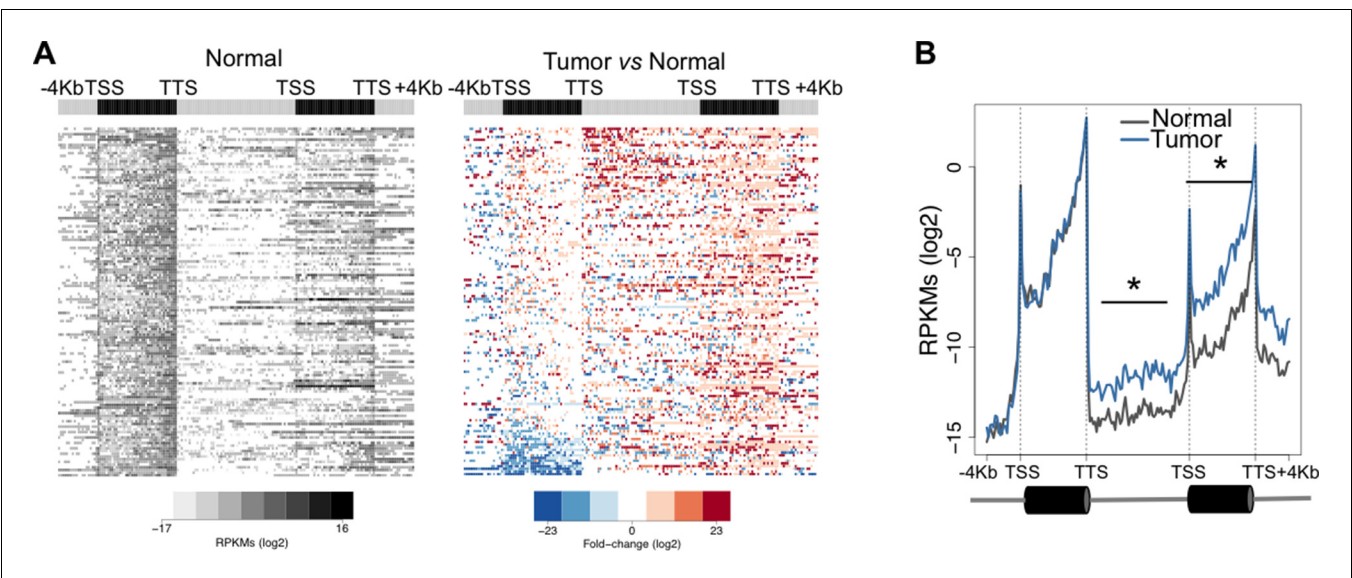

**Figure 5.** Transcription read-through overruns and interferes with the expression of neighbouring genes. Heatmap (**A**) and metagene (**B**) profiles of the intergenic region and of the gene located downstream of a read-through event in one representative ccRCC TCGA sample (patient barcode TCGA-CZ-5465). Genes were scaled and averaged as in *Figure 1*. *p<0.05 by Student's T-test.

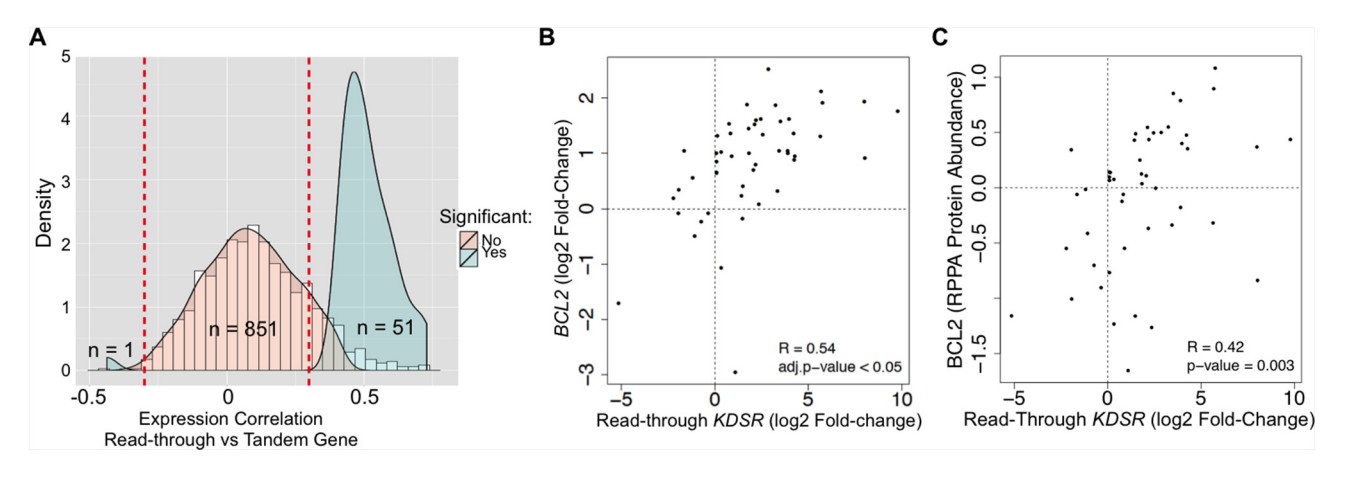

**Figure 6.** Transcription read-through of the KDSR gene correlates with the expression of the BCL2 oncogene. (**A**) Distribution of the correlation between the read-through and the expression levels of the downstream tandem genes. Significant correlation values (Benjamini-Hochberg adjusted p<0.05) are represented in blue. (**B**) Correlation between the expression levels of *BCL2* and the read-through of the upstream *KDSR* gene. The graph depicts the fold-change of read counts for each tumor and matched normal pair. (**C**) Correlation between BCL2 protein levels and the *KDSR* read-through.

for the aberrant expression of cancer-related genes, which may explain the upregulation of the *BCL2* oncogene in ccRCC.

## Transcription read-through is a source of RNA chimeras in ccRCC

RNAPII elongation beyond the annotated gene boundaries and invasion of an adjacent gene as a result of impaired transcription termination may result in the formation of hybrid transcripts collectively called RNA chimeras (*Gingeras, 2009*). In addition to gene fusions (formed upon chromosomal translocations) RNA chimeras can originate from DNA-independent events such as *trans*-splicing, RNA recombination or transcriptional read-through (*Gingeras, 2009*). Our analysis of ccRCC RNA-seq data revealed a high number of reads mapping at two distinct tandem genes, which are indicative of RNA chimeras generated by intergenic splicing following a read-through event (*Figure 7A*, *Supplementary file 1H*). Analysis of the splicing pattern of these chimeras showed that most events join the second-last exon of the upstream gene with the second exon of the downstream gene (*Figure 7B*). This pattern is compatible with the exon definition model according to which transcription termination is necessary for the selection of the terminal 3' splice site (ss) (*Niwa et al., 1992*; *Dye and Proudfoot, 1999*). In the absence of transcription termination, the terminal 3'ss is evicted and the terminal 5'ss will splice together with the first 3'ss of the downstream gene, which emerges from the nascent transcript once RNAPII reaches the second exon (*Figure 7B*). Interestingly, the number of RNA chimeras detected correlated positively with the levels of transcription read-through (p = 0.003; R = 0.41) and was significantly higher in the sample group with a 'high read-through' phenotype defined above (*Figure 7C*).

A remarkable feature of these RNA chimeras is that some were recurrently detected in different tumor samples. One particular chimera, encoded by two consecutive genes - *CTSC* and *RAB38* - was detected in 20% of the TCGA samples (but not in any matched normal sample) and in four of the six ccRCC cell lines that we sequenced de novo (*Supplementary file 1H*). We validated this RNA chimera by RT-qPCR before and after transfection of the ccRCC cell lines with a small interfering RNA (siRNA) spanning the transcript break-point, which resulted in a robust depletion of *CTSC-RAB38* (*Figure 8A,B*). In contrast, siRNAs targeting either the last exon of *CTSC* or the first exon of *RAB38* (which are not included in the chimeric transcript) significantly decreased the levels of *CTSC* and *RAB38*, respectively, but not the levels of the *CTSC-RAB38* chimera (*Figure 8B,C*). Moreover, we measured the RNAPII occupancy throughout the intergenic region and within the body of the *CTSC* and *RAB38* genes by chromatin immunoprecipitation (ChIP). We detected a robust occupancy of RNAPII throughout the intergenic region (*Figure 8D*), which further suggests that the *CTSC-RAB38*

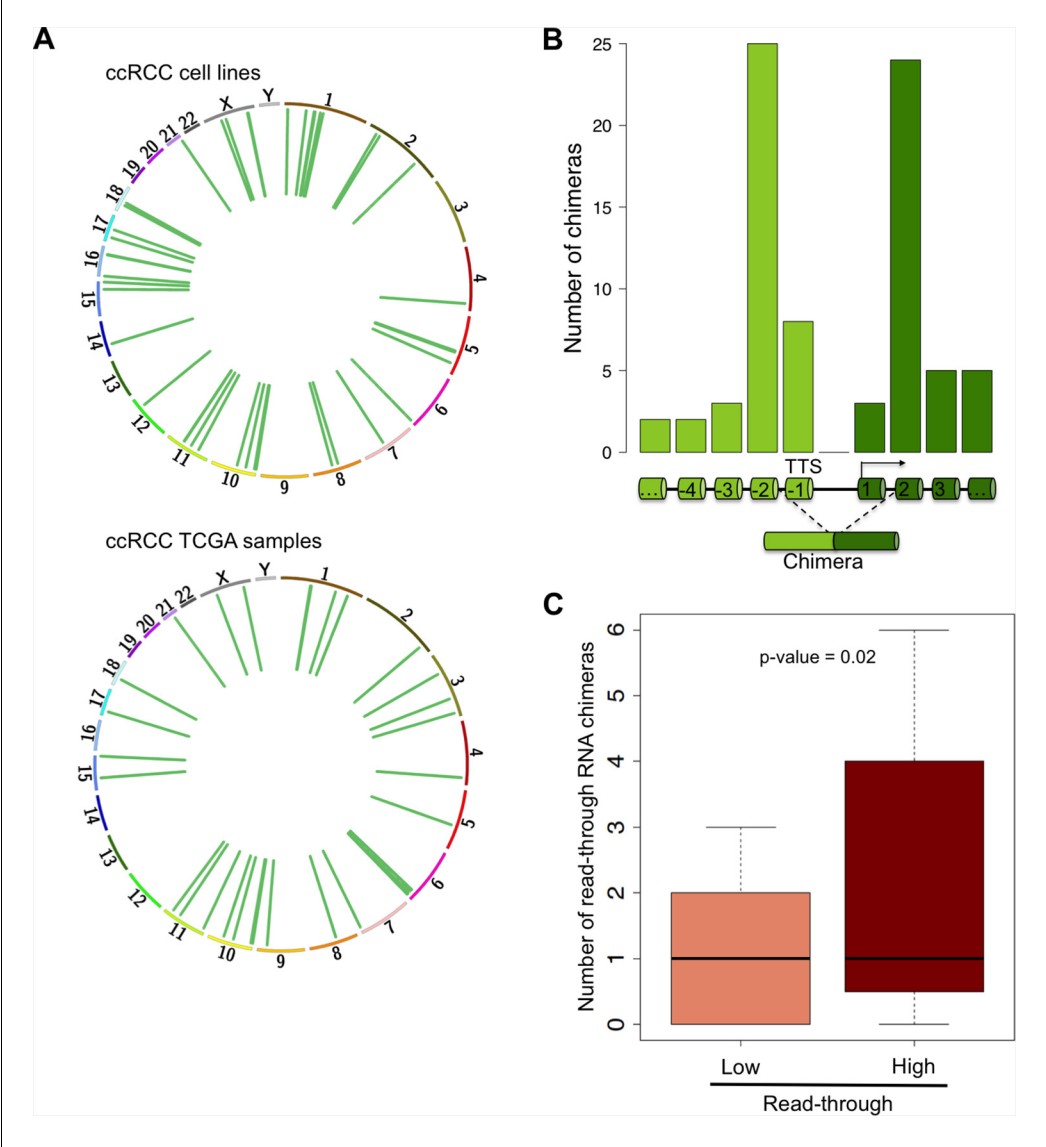

**Figure 7.** Read-through RNA chimeras are prevalent in ccRCC. (**A**) Circos plots showing the location of genes forming each RNA chimera detected on human ccRCC cell lines and TCGA samples. Chimeras are represented by curves inside the Circos. (**B**) Number of read-through RNA chimeras formed by intergenic splicing between the represented exons. (**C**) Number of read-through RNA chimeras in the low and high read-through sample subsets defined in *Figure 2A*.

chimera is generated following a read-through episode without any genomic deletion or transloca-tion involved.

## Discussion

Our transcriptome analysis of a large dataset of tumor and normal matched samples revealed that transcription events extending beyond the annotated 3' end of genes are frequent in ccRCC. Strik-ingly, our analysis further disclosed an unexpected prognostic power of transcription read-through in kidney cancer: higher number of genes with transcription read-through correlates significantly with poorer patient survival. Amongst the most frequently mutated genes in ccRCC, we identified *SETD2* inactivation as a contributing factor for increased transcription read-through. In fact, ectopic expression of *SETD2* was sufficient to rescue the transcription termination defects of *SETD2* mutant ccRCC cells. Moreover, we report that the effects of impaired transcription termination are not

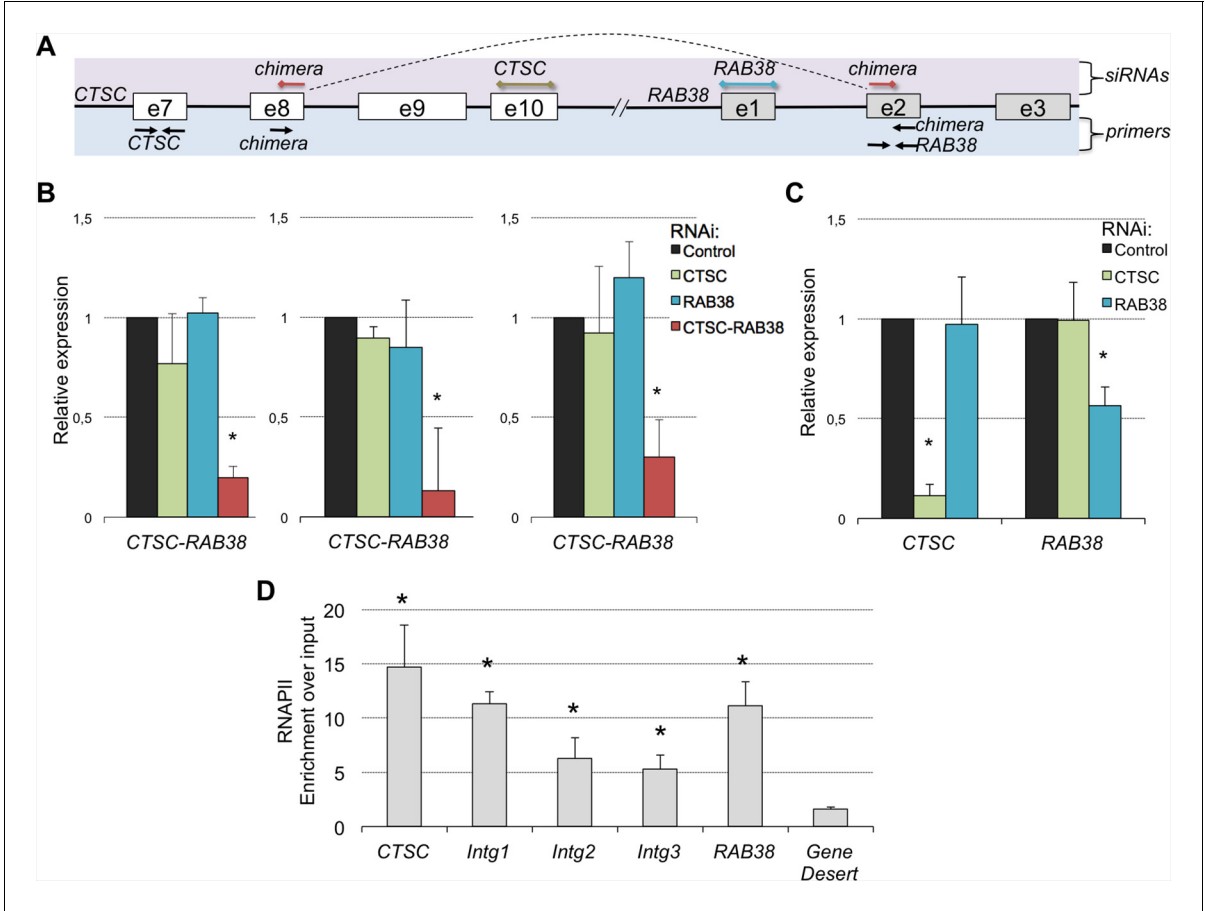

**Figure 8.** The CTSC-RAB38 chimera is recurrently detected in ccRCC. (**A**) Schematic illustration of the *CTSC-RAB38* locus depicting the position of the primers used to measure the transcripts levels by RT-qPCR shown in (**B**) and (**C**) and the position of the siRNAs targeting each of the three transcripts (*CTSC, RAB38* and the *CTSC-RAB38* chimera). The dashed curve in the scheme illustrates the splicing pattern of the chimeric transcript. (**B**) Relative expression of the *CTSC-RAB38* chimeric transcript after depletion of the indicated transcripts by RNAi in three distinct ccRCC cell lines (FG2, MF, ER). (**C**) Relative expression of *CTSC* and *RAB38* upon depletion of the indicated transcripts by RNAi in FG2 cells. Similar results were obtained with the other ccRCC cell lines. *p<0.05 by Student's T-test compared to controls. (**D**) RNAPII ChIP along the *CTSC-RAB38* locus in FG2 cells. Means and standard deviations from five independent experiments are shown. *p<0.05 by Student's T-test compared to the gene desert.

confined to the affected gene. Instead, it also contaminates the expression of neighboring genes that are overran by reading-through RNAPII complexes. Importantly, amongst the genes whose expression directly correlates with the volume of transcription read-through on the upstream gene, we detected the anti-apoptotic oncogene *BCL2*. This finding unveils a new source of aberrant expression of cancer-related genes and provides a plausible mechanistic basis for the upregulation of *BCL2* frequently observed in ccRCC.

Our present study further reveals recurrent RNA chimeras in ccRCC combining sequences from two tandem genes. Such chimeras are generated following extensions of RNAPII beyond the annotated gene boundaries and invasion of an adjacent gene as a result of impaired transcription termination. According to the exon definition model (*Niwa et al., 1992*; *Dye and Proudfoot, 1999*), deficient transcription termination is expected to impair the splicing of the last exon. In agreement, most chimeras skip the last exon of the upstream gene and the prevailing splicing pattern joins the second-last exon of the upstream gene with the second exon of the downstream gene. The finding that several of these RNA chimeras are recurrently detected in two or more tumor samples, suggests that they were selected during cancer development and supports the exciting possibility that they may play relevant functional roles.

Although our study primarily focused on the characterization of the transcription termination defects and their impact on the ccRCC transcriptome, our pioneer findings raise intriguing questions such as: which mechanism(s) drive such transcription defects?; how individual RNA chimeras are functionally involved in tumorigenesis?; do chimeric transcripts produce functional oncoproteins and/or can they directly affect the expression of other relevant cancer genes?; how can we intervene therapeutically to restore the canonical transcription pattern? The widespread incidence of aberrant termination in ccRCC cells suggests that it may play important roles in expanding the transcriptome diversity that drives cancer development and progression. The impact on *BCL2* expression illustrates a relevant functional outcome of transcription read-though. The generation of chimeric transcripts, namely those recurrently identified in several samples, such as *CTSC-RAB38*, further discloses the contribution of impaired transcription termination for the expansion of RNA species that may favor cancer progression. Nevertheless, future studies should provide direct evidence that a chimeric transcript has oncogenic functions in order to support the physiological relevance of these RNAs. This is a challenging and very exciting topic and further efforts are required to fully elucidate the impact of impaired transcription termination on cancer in general and on ccRCC in particular.

## Materials and methods

### Cell culture
ccRCC cells (Caki-1, Caki-2, MF, ER, AB and FG2, Cell Line Services Eppelheim, Germany) were grown as monolayers in Dulbecco's modified Eagle medium (DMEM, Invitrogen, CA, USA), supplemented with 10% (v/v) FBS, 1% (v/v) nonessential amino acids, 1% (v/v) L-glutamine and 100U/ml penicillin-streptomycin, and maintained at 37°C in a humidified atmosphere with 5% $CO_2$.

### RNA interference
RNAi was achieved using synthetic siRNA duplexes (Eurogentec, Belgium). The sequence of the siRNAs is shown in *Supplementary file 1I*. siRNAs targeting the firefly luciferase (GL2) were used as controls. Cells were reverse transfected with 10 µM siRNAs using OptiMEM (Invitrogen) and Lipofectamine RNAiMAX (Invitrogen), according to the manufacturer's instructions. 24 hr after the first transfection, cells were re-transfected with the same siRNA duplexes and transfection reagents and harvested on the following day.

### RNA isolation and quantitative RT–PCR
Total RNA was extracted with TRIzol (Invitrogen). cDNA was made using Superscript II Reverse Transcriptase (Invitrogen). RT-qPCR was performed in the ViiA Real Time PCR (Applied Biosystems, CA, USA), using SYBR Green PCR master mix (Applied Biosystems). The relative RNA expression was estimated as follows: $2^{(Ct\ reference\ -\ Ct\ sample)}$, where Ct reference and Ct sample are mean threshold cycles of RT-qPCR done in duplicate on cDNA samples from *U6 snRNA* (reference) and the cDNA from the genes of interest (sample). All primer sequences are presented in *Supplementary file 1I*.

### Chromatin immunoprecipitation
ChIP was performed as described (*de Almeida et al., 2011*). The relative occupancy of RNAPII at each locus was estimated by RT-qPCR as follows: $2^{(Ct\ Input\ -\ Ct\ IP)}$, where Ct Input and Ct IP are mean threshold cycles of RT-qPCR done in duplicate on DNA samples from input and specific immunoprecipitations, respectively. RNAPII was precipitated with an antibody against its largest subunit (N20, sc-899; Santa Cruz, TX, USA). The sequences of gene-specific, intergenic regions and gene desert primer pairs are presented in *Supplementary file 1I*.

### RNA fluorescence in situ hybridization (FISH)
FG2 ccRCC cells were transiently transfected with a wt *SETD2-GFP* expression plasmid (*Carvalho et al., 2014*) and cultured on glass coverslip for 24 hr before hybridization with RNA FISH probes (Biosearch Technologies, CA, USA) following the manufacturer's protocol. Briefly, cells were washed with PBS, fixed for 10 min at room temperature, washed twice with PBS, and permeabilized at 4°C in 70% (vol/vol) EtOH. Probes diluted in hybridization buffer were added to permeabilized

cells before overnight incubation in a dark chamber at 37°C. After washing, DAPI was added to stain the nuclei. Epi-fluorescence microscopy was performed using a Zeiss Z1 microscope equipped with Z-piezo (Prior, MA, USA), a 63x 1.4 NA Plan-Apochromat objective and a sCMOS camera (Hamamatsu Flash 4.0). FISH probes were designed to target a segment of the RNA transcript encoded by the intergenic region downstream of the canonical termination sites of *MRPL23* and *SEL1L3*. The sequences of the probes are shown in *Supplementary file 1J*.

## RNA-seq datasets and preprocessing

Samples were barcoded and prepared for sequencing by the Centro Nacional de Análisis Genómico (CNAG, Barcelona, Spain) using Illumina protocols. PolyA+ RNA-seq libraries of ccRCC cell lines were sequenced as paired-end 75-bp sequence tags using the standard Illumina pipeline. ccRCC RNA-seq datasets (with at least 40 million mapped reads on each tumor and matched normal samples – in a total of 50 paired-samples listed in *Supplementary file 1A*) were obtained from TCGA. HEK293 RNA-seq data were from the Sequence Read Archive (SRX876600). RNA-seq data from *SETD2* KO ccRCC cells (786-O), *SETD2*-depleted human MS cells and *Setd2* KO mouse ES cells were obtained from the GEO (http://www.ncbi.nlm.nih.gov/geo/ GSE66879, GSE19373 and GSE54932, respectively). Data quality was assessed with the FastQC (http://www.bioinformatics.babraham.ac.uk/projects/fastqc/) quality-control tool for high throughput sequence data. Sequence tags were then mapped to the reference human (hg19) or mouse (mm9) genomes with TopHat software using default parameters (*Kim et al., 2013*). Reads from samples with multiple sequencing lanes were merged and only the best score alignment was accepted for each read.

## Transcriptome alterations

Gene annotations were obtained from UCSC knownGene and refGene tables (*Karolchik et al., 2014*) and merged into a single transcript model per gene using BedTools (*Quinlan and Hall, 2010*). Our analysis was restricted to transcriptionally active genes defined as those with expression levels higher than the 25th percentile. To identify genes with transcriptional read-through, we filtered out all genes for which there was another annotated gene in either strand within a region of 5 Kb downstream of their TTS. We also filtered out genes with an overall increase in expression level relative to the control sample. Reads were counted in 100 bp windows for the 4 Kb region downstream of the TTS and normalized for the total number of mapped reads on each sample (RPKMs) (*Mortazavi et al., 2008*). We considered the occurrence of transcriptional read-through when more than six 100 bp windows showed increased (at least 1.5 fold-change) RPKMs relative to control. The control samples were: Caki2 for ccRCC cell lines; the matched normal tissue sample for each ccRCC patient; control 786-O ccRCC cells for *SETD2* KO 786-O cells; human MS cells for *SETD2*-depleted human MS cells; and wt mouse ES cells for the *Setd2* KO mouse. In the ccRCC cell lines dataset, all the analysis were also performed comparing both *SETD2* wt (Caki1 vs Caki2), with this comparison working as a negative control. Statistical significance of the differences between proportions of genes showing read-through was assessed using the Fisher's exact test. RNA chimeras were detected using the fusion-search option (fusion-min-dist set to 100 bp) in TopHat alignment (*Kim et al., 2013*) and TopHat-Fusion (*Kim and Salzberg, 2011*) with default parameters. For downstream analysis we only considered RNA chimeras supported by at least two reads. A set of in-house scripts were written in bash and in the R environmental language (http://www.R-project.org/) (*Team RDC, 2011*).

## Graphical representation of data

Figures were produced using BedTools (*Quinlan and Hall, 2010*) and default packages from the R environment. To produce heatmaps and metagene average profiles showing transcription read-through, genes were scaled to 60 equally sized bins so that all annotated TSSs and TTSs were aligned. Regions 4 Kb upstream of TSSs and 4 Kb downstream of TTSs were averaged in a 100-bp window. Individual gene profiles were produced by successions of 10-bp windows (single gene) or 100-bp windows (two genes). All read counts were normalized by genomic region length and number of mapped reads (RPKM), and RPKM values were log2 transformed when representing multiple genes. Protein levels assessed by reverse phase protein array (RPPA) were gathered from the TCGA

portal. Circle plots showing RNA chimeras distribution were produced using Circos (*Krzywinski et al., 2009*).

## Accession codes for RNA-seq data

The RNA-seq data for ccRCC cell lines have been deposited in Gene Expression Omnibus (GEO; http://www.ncbi.nlm.nih.gov/geo/) database under the accession number GSE64451. ccRCC RNA-seq datasets were obtained from TCGA. HEK293 RNA-seq data were from the Sequence Read Archive (SRX876600). RNA-seq data from *SETD2* KO 786-O ccRCC cells, human mesenchymal stem cells and mouse embryonic stem cells were obtained from the GEO (GSE66879, GSE19373 and GSE54932, respectively).

## Acknowledgements

We thank our colleagues Nuno Barbosa-Morais, Sérgio Dias and Edgar Gomes for critical comments and suggestions. We also thank Ioana Posa, Mafalda Pimentel, Célia Carvalho and the Bioimaging Unit of the IMM for technical assistance. This work was supported by Fundação para a Ciência e Tecnologia (FCT), Portugal (PTDC/BIM-ONC/0384-2012 to SFdA). MRM is a FCT PhD fellow (SFRH/BD/92208/2013). ACV is a Lisbon BioMed PhD fellow funded by FCT (SFRH/BD/52232/2013). ARG is the recipient of a FCT Investigator award (IF/00510/2014).

## Additional information

### Funding

| Funder | Grant reference number | Author |
|---|---|---|
| Fundação para a Ciência e a Tecnologia | PTDC/BIM-ONC/0384-2012 | Sérgio F de Almeida |
| Fundação para a Ciência e a Tecnologia | SFRH/BD/92208/2013 | Mafalda R Matos |
| Fundação para a Ciência e a Tecnologia | SFRH/BD/52232/2013 | Alexandra C Vítor |
| Fundação para a Ciência e a Tecnologia | IF/00510/2014 | Ana R Grosso |

The funders had no role in study design, data collection and interpretation, or the decision to submit the work for publication.

### Author contributions

ARG, SFDA, Conception and design, Acquisition of data, Analysis and interpretation of data, Drafting or revising the article; APL, Conception and design, Acquisition of data, Analysis and interpretation of data; SC, MRM, FBM, ACV, Acquisition of data, Analysis and interpretation of data; JMPD, Acquisition of data, Analysis and interpretation of data, Contributed unpublished essential data or reagents; MCF, Analysis and interpretation of data, Drafting or revising the article

### Author ORCIDs

Ana P Leite, http://orcid.org/0000-0001-5773-3211
Sérgio F de Almeida, http://orcid.org/0000-0002-7774-1355

## Additional files

### Supplementary files

• Supplementary file 1. (A) Genes with transcription read-through identified in human ccRCC TCGA samples. (B) Genes involved in transcription termination and pre-mRNA processing with mutations in TCGA samples. (C) Genes with transcription read-through identified in human ccRCC cell lines. (D) Genes with transcription read-through identified in human 786-O ccRCC cells. (E) Genes with transcription read-through identified in human mesenchymal stem cells. (F) Genes with

transcription read-through identified in mouse embryonic stem cells. (G) Genes with transcription read-through identified in human ccRCC TCGA samples, showing significant correlation between read-through levels and expression of downstream gene. (H). Read-Through RNA chimeras identified in human ccRCC TCGA samples and ccRCC cell lines. (I). siRNAs and Primer Sequences. (J) Sequences of RNA FISH probes.

#### Major datasets

The following datasets were generated:

| Author(s) | Year | Dataset title | Dataset URL | Database, license, and accessibility information |
|---|---|---|---|---|
| Grosso AR, Leite AP, Carvalho S, Matos M, Martins FB, Vitor AC, Desterro J, Carmo-Fonseca M, de Almeida SF | 2015 | Pervasive transcription read-through promotes aberrant expression of oncogenes and RNA chimeras in renal carcinoma | http://www.ncbi.nlm.nih.gov/geo/query/acc.cgi?acc=GSE64451 | Publicly available at the NCBI Gene Expression Omnibus (Accession no: GSE64451). |

The following previously published datasets were used:

| Author(s) | Year | Dataset title | Dataset URL | Database, license, and accessibility information |
|---|---|---|---|---|
| Luco RF, Pan Q, Tominaga K, Blencowe BJ, Pereira-Smith OM, Misteli T | 2010 | Regulation of alternative splicing by histone modifications | http://www.ncbi.nlm.nih.gov/geo/query/acc.cgi?acc=GSE19373 | Publicly available at the NCBI Gene Expression Omnibus (Accession no: GSE19373). |
| Zhang Y, Chen Z, Huang Q, Chen S | 2014 | Quantitative Analysis of Wild Type (wt) and Setd2 knockout(ko) mESCs Transcriptomes [RNA-Seq] | http://www.ncbi.nlm.nih.gov/geo/query/acc.cgi?acc=GSE54932 | Publicly available at the NCBI Gene Expression Omnibus (Accession no: GSE54932). |
| Memorial Sloan Kettering Cancer Center | 2015 | RNA-seq of HEK293 with empty vector (replicate 2) | http://www.ncbi.nlm.nih.gov/sra/?term=SRX876600 | Publicly available at the NCBI Short Read Archive (Accession no: SRX876600). |
| Ho TH, Nie J, Yan H | 2015 | RNA sequencing of SETD2 isogenic renal cell carcinoma cell lines | http://www.ncbi.nlm.nih.gov/geo/query/acc.cgi?acc=GSE66879 | Publicly available at the NCBI Gene Expression Omnibus (Accession no: GSE66879). |

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
