## [Decision Letter]

Thank you for submitting your work entitled "Pervasive transcription read-through promotes aberrant expression of oncogenes and RNA chimeras in renal carcinoma" for peer review at *eLife*. Your submission has been favorably evaluated by Mark McCarthy (Senior editor) and three reviewers, one of whom, Chi Dang, is a member of our Board of Reviewing Editors.

The reviewers have discussed the reviews with one another and the Reviewing editor has drafted this decision to help you prepare a revised submission.

Summary:

The manuscript titled, "Pervasive transcription read-through promotes aberrant expression of oncogenes and RNA chimeras in renal carcinoma" by Grosso et al., analyzes the transcription profiles from publically available database (TCGA) of RNA-seq results from 50 ccRCC primary samples to identify improper termination events that occur in these samples. The authors identified transcriptional read-through (TRT) events in the entire genome. While this effect is noticeable it is also subtle as summarized in Figure 1. Even though the effect occurs not exclusively in those tumors where *SetD2* is mutated, grouping the tumor samples according to *SetD2* mutations shows that these have an overall higher prevalence of TRT, which also correlates with higher read numbers of the adjacent gene and occurrence of chimeric transcripts. The authors conclude that mutations in the *SETD2* gene, which is involved in setting H3K36 methylation along transcribed sequences, are a common feature of the ccRCC and the cause of TRT.

The same group reported previously that mutations in *SETD2* are linked to reduced double-strand break repair and activation of the p53 pathway. In the current manuscript, they document a significant level of TRT in RCC, in which increased *BCL2* gene expression is associated with TRT into the body of the *BCL2* gene. In addition, chimeric transcripts also arise from TRT, and the fusion transcript *CTSC-RAB38* is found in 20% of RCC. Using *SETD2* knockout cells, the authors also documented an increase in TRT as compared with wild-type cells. Further, the authors report the formation of RNA chimeras resulting from the improper splicing of the chimeric RNAs resulting from the transcriptional read-through.

Overall, this manuscript is interesting, although their main focus, as suggested in the title, of RNA chimeras has been reported previously for ccRCC in the Nature paper published a few years ago (Nature 499,43-49 (04 July 2013) doi:10.1038/nature12222). Furthermore, a recent study in neuroblastoma revealed pervasive TRT in response to osmotic shock, which was seemingly dependent on IP3R activation (Vilborg et al. Mol Cell 2015).

Essential revisions:

While the technical aspect of this study is very well-done and the findings offer a different conceptual framework for the cancer transcriptome, they are only based on correlations and it remains unclear if they contribute to malignancy and are mechanistically linked to loss of *SetD2*. Without further functional evidence that the absence of *SETD2* directly accounts for enhanced read-through the current work remains too preliminary. This could be achieved by knockdown of *SETD2* in wild-type RCC cell lines followed by TRT measurement to determine whether loss of *SETD2* function would reduce TRT. Using inducible *SETD2* in *SETD2* mutant RCC cell lines, one could potentially perform short term TRT measurement upon induction of *SETD2* expression and/or provide colony suppression assays to document the inability to reconstitute *SETD2* expression in mutant cell lines.

Minor points:

1) The authors claim that *SETD2* mutations cause the improper readout using Caki-2 cell lines as a control. The Caki-2 cell line is also a ccRCC cell line, and the other lines used in this study show lower levels of *Setd2* transcription. However, the authors should show that mutations in *BAP, PBRM1* or *VHL* do not show any transcriptional read-through before arriving at that conclusion. It would be informative if the authors compared the levels of *Setd2* with a cell line like HEK293, to get an idea as to how high it is expressed in Caki-2 cells relative to a 'normal' cell line.

2) The authors do not discuss or offer any explanations as to why the RNA chimera skips the last exon of the preceding gene and the first exon of the following gene. A comment on that observation would be interesting in the Discussion.

3) The authors should discuss the following points. Further work will be necessary to document whether chimeric transcripts produce functional oncoproteins and whether TRT results in transcripts that could increase the production of downstream genes, such as *BCL2*. The latter could be studied in future work through ribosome profiling and footprinting. Furthermore, direct evidence that a chimeric transcript (since it is unclear if these are indeed translated) has oncogenic functions will have to be studied in the future to justify the proposed model.

---

## [Author Response]

Essential revisions:

*While the technical aspect of this study is very well-done and the findings offer a different conceptual framework for the cancer transcriptome, they are only based on correlations and it remains unclear if they contribute to malignancy and are mechanistically linked to loss of* SetD2*. Without further functional evidence that the absence of* SETD2 *directly accounts for enhanced read-through the current work remains too preliminary. This could be achieved by knockdown of* SETD2 *in wild-type RCC cell lines followed by TRT measurement to determine whether loss of* SETD2 *function would reduce TRT.*

Following the reviewers’ request, we now provide direct measurements of transcription read-through upon *SETD2* depletion in wild-type (wt) ccRCC cells. These were achieved using two distinct approaches: (a) Analysis of RNA-seq data from *SETD2* wt and *SETD2*-knockout ccRCC cells (Figure 3) and (b) RT-qPCR analysis of transcription read-through in three different genes (*MRPL23, SEL1L3 and FAM46A)* following RNAi-mediated knockdown of *SETD2* in a wt ccRCC cell line (Figure 3—figure supplement 3). These additional data further support our conclusion that *SETD2* is necessary to prevent transcription read-through.

*Using inducible* SETD2 *in* SETD2 *mutant RCC cell lines, one could potentially perform short term TRT measurement upon induction of* SETD2 *expression and/or provide colony suppression assays to document the inability to reconstitute* SETD2 *expression in mutant cell lines. Without these experiments, the manuscript would not be acceptable for publication.*

To evaluate if wt *SETD2* can rescue the transcription read-through phenotype, we performed single-molecule RNA FISH upon transient transfection of mutant ccRCC cell lines with wt *SETD2*. We used FISH probes designed against a segment of the RNA transcript encoded by the intergenic region downstream of the canonical termination site of two different genes (*MRPL23* and *SEL1L3*, Figure 4). These new data show that reconstitution of *SETD2* activity in mutant ccRCC cells rescues the transcription termination defects.

*Minor points:1) The authors claim that* SETD2 *mutations cause the improper readout using Caki-2 cell lines as a control. The Caki-2 cell line is also a ccRCC cell line, and the other lines used in this study show lower levels of* Setd2 *transcription. However, the authors should show that mutations in* BAP, PBRM1 *or* VHL *do not show any transcriptional read-through before arriving at that conclusion. It would be informative if the authors compared the levels of* Setd2 *with a cell line like HEK293, to get an idea as to how high it is expressed in Caki-2 cells relative to a 'normal' cell line.*

As we show in Figure 2, ccRCC samples with mutations in *BAP, PBRM1* or *VHL* do have transcription read-through. However, we show that mutations in *SETD2* correlate significantly with higher levels of read-through. In the revised manuscript we now present new data showing that loss of *SETD2* is sufficient to cause widespread transcription read-through in a wt ccRCC cell line. Moreover, we show that ectopic expression of *SETD2* in a mutant ccRCC cell line rescues the termination defects. In addition, following the referees’ suggestion, we now include HEK293 cells in the comparative analysis shown in Figure 3. These new data reveal that HEK293 and Caki-2 cells express *SETD2* at similar levels.

2) The authors do not discuss or offer any explanations as to why the RNA chimera skips the last exon of the preceding gene and the first exon of the following gene. A comment on that observation would be interesting in the Discussion.

We show in Figure 7 that the splicing pattern of the chimeras join the second-last exon of the upstream gene with the second exon of the downstream gene. We suggest that this pattern is in agreement with the exon definition model according to which the absence of transcription termination evicts the last 3’ss of the upstream gene favouring the use of the first 3’ss that emerges from the downstream gene. This model explains why the RNA chimeras skip the last exon of the upstream gene and the first exon of the downstream gene. We now comment this model in the Discussion.

*3) The authors should discuss the following points. Further work will be necessary to document whether chimeric transcripts produce functional oncoproteins and whether TRT results in transcripts that could increase the production of downstream genes, such as* BCL2*. The latter could be studied in future work through ribosome profiling and footprinting. Furthermore, direct evidence that a chimeric transcript (since it is unclear if these are indeed translated) has oncogenic functions will have to be studied in the future to justify the proposed model.*

These points are now discussed in the revised manuscript.